# If LLM Is the Wizard, Then Code Is the Wand: A Survey on How Code Empowers Large Language Models to Serve as Intelligent Agents

## Abstract

The prominent large language models (LLMs) of today differ from past language models not only in size, but also in the fact that they are trained on a combination of natural language and formal language (code). As a medium between humans and computers, code translates high-level goals into executable steps, featuring standard syntax, logical consistency, abstraction, and modularity. In this survey, we present an overview of the various benefits of integrating code into LLMs' training data. Specifically, beyond enhancing LLMs in code generation, we observe that these unique properties of code help *i)* unlock the reasoning ability of LLMs, enabling their applications to a range of more complex natural language tasks; *ii)* steer LLMs to produce structured and precise intermediate steps, which can then be connected to external execution ends through function calls; and *iii)* take advantage of code compilation and execution environment, which also provides diverse feedback for model improvement. In addition, we trace how these profound capabilities of LLMs, brought by code, have led to their emergence as intelligent agents (IAs) in situations where the ability to understand instructions, decompose goals, plan and execute actions, and refine from feedback are crucial to their success on downstream tasks. Finally, we present several key challenges and future directions of empowering LLMs and IAs with code.

## 1 Introduction

Code has become an integral component in the training data of large language models (LLMs), including well-known models such as Llama2, GPT-3.5 series and GPT-4 (Touvron et al., 2023; Ye et al., 2023a; OpenAI, 2023). Training LLMs on code has gained popularity not only because the acquired programming skills enable commercial applications, such as Github Copilot[1], but also because it improves the models' previously lacking reasoning abilities (Liang et al., 2023b). Consequently, LLMs rapidly emerge as a primary decision-making hub for intelligent agents (IAs) (Zhao et al., 2023), demonstrating an exponential growth in capabilities from code training and the advancement of tool learning (Qin et al., 2023). These LLM-based IAs are poised to handle a wider range of more complex tasks, including downstream applications in multi-agent environment simulation (Wu et al., 2023b) and AI for science (Boiko et al., 2023).

As depicted in Figure 1, this survey aims to explain the widespread adoption of code-specific training in the general LLM training paradigm and how code enhances LLMs to act as IAs. Unlike previous code-LLM surveys that concentrate on either evaluating and comparing code generation abilities (Zan et al., 2023; Xu et al., 2022), or listing IA tasks (Wang et al., 2023d; Xi et al., 2023; Zhao et al., 2023) in IA surveys, we aim to provide a comprehensive understanding of how code assists LLMs and where code benefits LLMs as IAs, based on the taxonomy of relevant papers (see Figure 2).

We first provide our definition of code and present typical methods for LLM code training (§2). Compared to natural language (refer to the case study in A.1), code is more structured, featuring logical, step-by-

---

[1]https://github.com/features/copilot

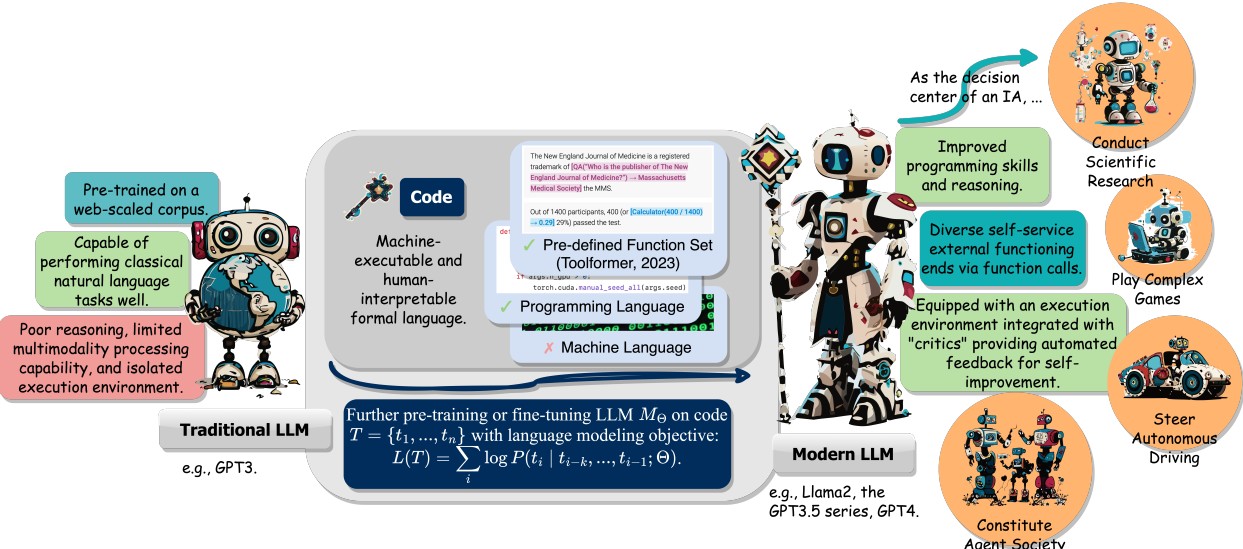

Figure 1: An illustration of how code empowers large language models (LLMs) and enhances their downstream applications as intelligent agents (IAs). While traditional LLMs excel in conventional natural language tasks like document classification and question answering, further pre-training or fine-tuning LLMs with human-interpretable and machine-executable code serves as an additional power-up — akin to equipping wizards with mana-boosting wands. This significantly boosts their performance as IAs through intricately woven operational steps.

step executable processes derived from procedural programming, as well as explicitly defined, modularized functions, which compose graphically representable abstractions. Additionally, code is typically accompanied by a self-contained compilation and execution environment. With insights from these characteristics of code, our comprehensive literature review reveals that integrating code into LLM training *i)* enhances their programming and reasoning capabilities (§3); *ii)* enables the models to directly generate executable, fine-grained steps during decision-making, thereby facilitating their scalability in incorporating various tool modules through function calls (§4); and *iii)* situates the LLMs within a code execution environment, allowing them to receive automated feedback from integrated evaluation modules and self-improve (§5).

In addition, as LLMs are becoming key decision-makers for IAs in complex real-world tasks, our survey also explores how these advantages facilitate their functioning along this capacity (§6), in terms of *i)* enhancing IAs' decision-making in perception and planning skills (§6.1), *ii)* facilitating their execution through direct action primitive grounding and modular memory organization (§6.2), and *iii)* providing an interactive environment for self-correction and self-improvement (§6.3). Finally, we discuss several open challenges and promising future directions (§7).

## 2 Preliminaries

### 2.1 Our Definition of Code

We consider code as any formal language that is both machine-executable and human-interpretable. For instance, human-readable programming languages fall within the scope of our discussion, whereas low-level languages, such as machine language based on binary instructions, are excluded due to their lack of human interpretability. Additionally, pre-defined formal languages, such as function sets employed in WebGPT (Nakano et al., 2021), are included as they can be parsed and executed in a rule-based manner.

LLMs trained with expressions formulated within a defined set of symbols and rules (e.g., pre-defined function sets, mathematical deduction formula, etc.), i.e., formal languages, exhibit advantages akin to those trained

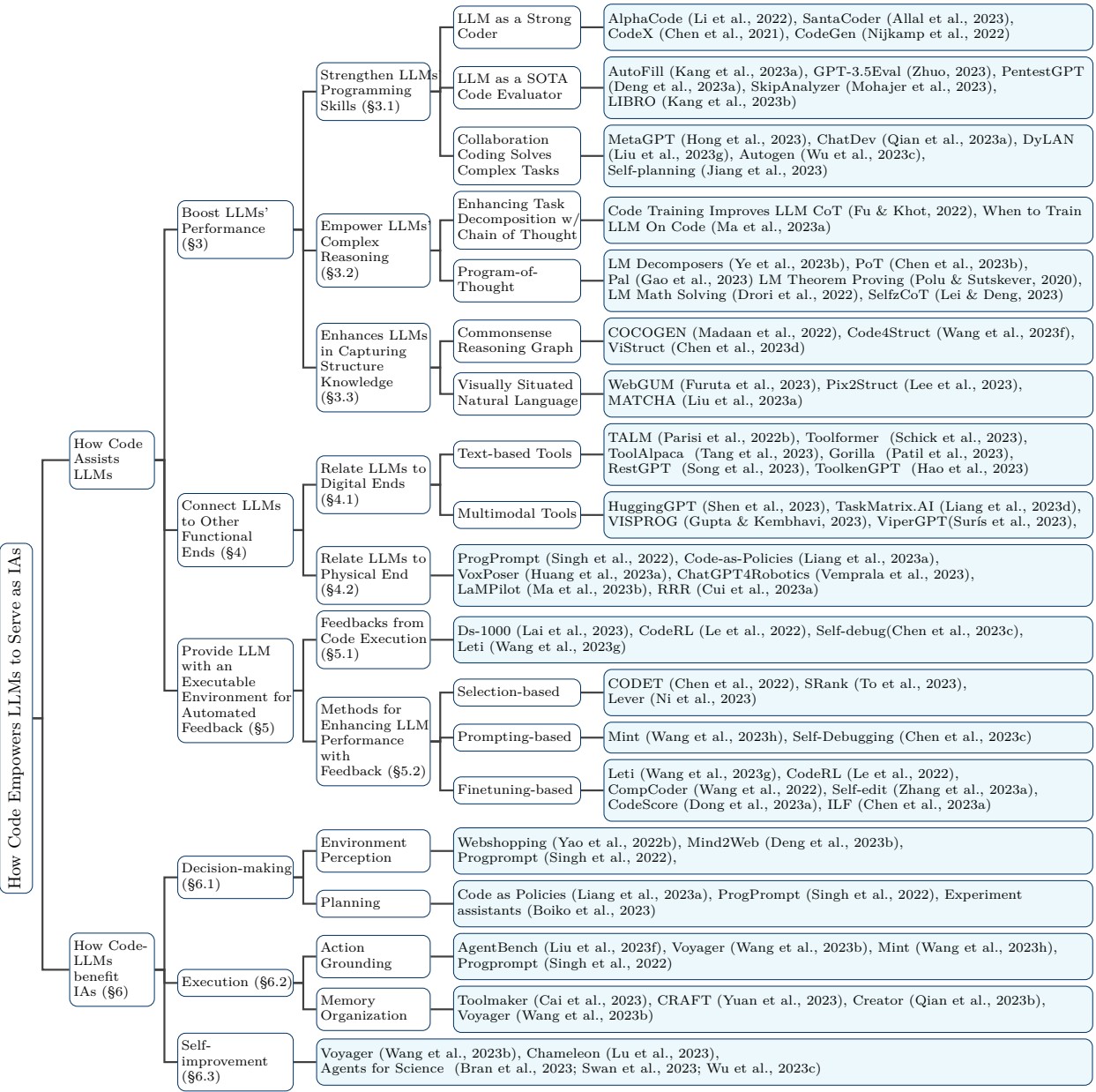

Figure 2: The organization of our paper, with a curated list of the most representative works. The complete work list is provided in Appendix D.

with programming languages. Therefore, we expand our definition of code to incorporate these homogeneous training corpora, enhancing the comprehensiveness of this survey to align with current research needs.

## 2.2 LLM Code Training Methods

LLMs undergo code training by following the standard language modeling objective, applied to code corpora. Given that code possesses natural language-like sequential readability, this parallels the approach to instruct LLMs in understanding and generating free-form natural language. Specifically, for an LLM $M_\Theta$ with parameters $\Theta$ and a code corpus $T = \{t_1, ..., t_n\}$, the language modeling loss for optimization is:

$$L(T) = \sum_i log P(t_i|t_{i-k}, ..., t_{i-1}; \Theta)$$

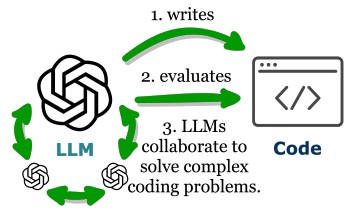
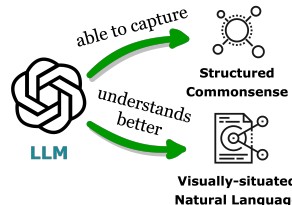

(a) Strengthen LLMs' programming and code evaluation skills (§3.1).

(b) Empower LLMs' complex reasoning, decoupling computation from language understanding (§3.2).

(c) Enable LLM to better capture structured knowledge and better understand complex multimedia data (§3.3).

Figure 3: How code pre-training boosts LLMs' performance.

When employing programming language (e.g., Python, C, etc.) as the corpus (Chen et al., 2021; Li et al., 2022; Nijkamp et al., 2022), training data is typically sourced from publicly accessible code repositories, such as GitHub. This process yields a corpus with a volume comparable to that of natural language pre-training, and thus we call training with such an abundance of code as *code pre-training*. The training strategy entails either training code on a pre-trained natural language LLM, as exemplified by Codex (Chen et al., 2021), or training a LLM from scratch with a blend of natural language and code, as demonstrated by CodeLLM (Ma et al., 2023a).

Conversely, when utilizing other pre-defined formal language for training, the objective shifts to acquainting the model with the application of specific functions (Schick et al., 2023), mathematical proof formulas (Wu et al., 2022), SQL (Sun et al., 2023b), and similar constructs. As the dataset for this is smaller compared to the pre-trained natural language corpus, we refer to such training process as *code fine-tuning*. Researchers apply the language modeling loss to optimize LLMs during this process, similarly.

## 3 Code Pre-Training Boosts LLMs' Performance

The pre-training of LLMs on code, exemplified by OpenAI's GPT Codex (Chen et al., 2021), has broadened the LLMs' scope of tasks beyond natural language. Such models enable diverse applications, including generating code for mathematical theory (Wu et al., 2022), general programming tasks (Chen et al., 2021), and data retrieval (Sun et al., 2023b; Cheng et al., 2023). Code necessitates producing logically coherent, ordered sequences of steps essential for valid execution. Moreover, the executability of each step within code allows for step-by-step logic verification. Leveraging and embedding both these properties of code in pre-training has improved LLM chain-of-thought (CoT) performance across many conventional natural language downstream tasks (Lyu et al., 2023; Zhou et al., 2023a; Fu & Khot, 2022), indicating improved complex reasoning skills. Implicitly learning from code's structured format, code LLMs demonstrate further improved performance on commonsense structured reasoning tasks, such as those related to markup, HTML, and chart understanding (Furuta et al., 2023; Liu et al., 2023a).

In the following sections, our objective is to elucidate why training LLMs on code and employing code-based prompts enhance their performance on complex downstream tasks. Specifically, we highlight three key areas where pre-training on code have benefited LLMs: *i)* enhancing programming proficiency in §3.1, *ii)* empowering complex reasoning capabilities in §3.2, and *iii)* facilitating the capture of structured commonsense knowledge in §3.3, as shown in Figure 3.

### 3.1 Strengthen LLMs' Programming Skills

**LLM as a strong coder.** Earlier language models only generate domain-specific programs (Ellis et al., 2019) or restrict to one of the generic programming languages, such as Java or C# (Alon et al., 2020). Empowered by the increasing number of parameters and computing resources, recent LLM-based code generation models (such as AlphaCode (Li et al., 2022), CodeGen (Nijkamp et al., 2022), SantaCoder (Allal et al., 2023), PolyCoder (Xu et al., 2022)) could master more than 10 languages within the same model and

show unprecedented success. A well-known work is CodeX (Chen et al., 2021), with 12 billion parameters that reads the entire GitHub database and is able to solve 72.31% of challenging Python programming problems created by humans. Recent studies (Zan et al., 2023; Xu et al., 2022; Du et al., 2023; Vaithilingam et al., 2022; Wong et al., 2023; Fan et al., 2023) have provided systematic surveys and evaluations of existing code-LLMs.

With its strong code generation ability, LLMs benefit various applications that rely on code, such as database administration (Zhou et al., 2023b), embedded control (Liang et al., 2023a), game design (Roberts et al.), spreadsheet data analysis (Liu et al., 2023c), and website generation (Calò & De Russis, 2023).

**LLM as a state-of-the-art code evaluator.** On the other hand, LLMs themselves could be state-of-the-art evaluators (i.e., analyze and score) for human or machine-generated codes. Kang et al. (2023a) leverage LLM-based models for code fault localization, while Zhuo (2023) uses GPT-3.5 to evaluate the functional correctness and human preferences of code generation. In addition, Deng et al. (2023a) design a LLM-based penetration testing tool and find that LLMs demonstrate proficiency in using testing tools, interpreting outputs, and proposing subsequent actions. Two recent efforts (Li et al., 2023a; Mohajer et al., 2023) also utilize LLM for examining and analyzing source code without executing it. Furthermore, LLMs are used for automatic bug reproduction in Kang et al. (2023b) and vulnerable software evaluation in Noever (2023).

**Multi-LLM collaboration solves complex coding problems.** Though code LLMs, like GitHub Copilot, have shown unprecedented success, one LLM agent alone could fail in complicated scenarios requiring multiple steps. Luckily, collaborative coding among several role-specific LLM agents exhibits more accurate and robust performance towards complex tasks. Hong et al. (2023) incorporates human programming workflows as guides to coordinate different agents. Dong et al. (2023b) assigned three roles: analyst, coder, and tester to three distinct "GPT-3.5"s, which surpasses GPT-4 in code generation. Meanwhile, Qian et al. (2023a) designs a chat-powered software development process, assigning more than three roles to separate LLM agents. Other similar methods (Liu et al., 2023g; Talebirad & Nadiri, 2023; Wu et al., 2023c; Jiang et al., 2023) all employ multiple code-LLM agents or different phases of the same agent for code generation, software developments or leveraging generated intermediate codes for other general purpose tasks.

### 3.2 Empower LLMs' Complex Reasoning

**Code pre-training improves chain-of-thought performance.** Logically, many complex tasks can be divided into smaller easier tasks for solving. CoT prompting, where prompt inputs are designed with chains of reasoning, allows the LLM to condition its generation with further steps of reasoning, providing a direct approach to task decomposition (Wei et al., 2023). CoT has seen success in the decomposition of many tasks, such as planning (Huang et al., 2022b) and evidence-based question answering (Dua et al., 2022; Ye et al., 2023b).

While LLM CoT ability was originally mainly attributed to dramatically increased model sizes (Wei et al., 2022b), recent evidence compiled by Fu & Khot (2022) suggests that much of the performance improvements from CoT stems from its pre-training on code. For instance, when comparing different versions of GPT-3 (i.e., v1 vs. v5), LLMs not trained on code, such as GPT-3's text-davinci-001, see a small but substantial accuracy improvement of 6.4 % to 12.4 % with CoT on the mathematical reasoning task GSM8k (Cobbe et al., 2021). In contrast, LLMs pre-trained on code, such as GPT-3's text-davinci-002 and Codex (Chen et al., 2021), see a dramatic performance improvement arising from CoT, with a remarkable accuracy increase of 15.6% to 46.9% and 19.7% to 63.1% respectively. Supporting this hypothesis proposed by Fu & Khot (2022), Ma et al. (2023a) show that pre-training on code in small-sized LLMs (2.6B) (Zeng et al., 2021) enhances performance when using CoT, and even more remarkably that smaller code-pretrained LLMs outperform their larger non-code counterparts across many different tasks. Furthermore, their study indicates that incorporating a greater volume of code during the initial phases of LLM training significantly enhances its efficacy in reasoning tasks. Nevertheless, tempering expectations, it is possible that the discrepancy in CoT performance between LLMs with and without code pre-training diminishes as the size of the models decreases, as the accuracy gap between the small LLMs in Ma et al. (2023a) was less than 3% when evaluating CoT. Notably, both Fu & Khot (2022) and Ma et al. (2023a) show that pre-training on code improves LLM performance in both standard and CoT prompting scenarios across downstream tasks.

**Program-of-thought outperforms chain-of-thought.** Furthermore, in comparison to vanilla CoT methods, LLMs that first translate and decompose a natural language task into code (Chen et al., 2023b; Gao et al., 2023), typically termed program-of-thought (PoT) prompting or program-aided language model, see sizable gains in tasks that require disambiguation in both language and explicit longitudinal structure. This approach is especially effective in complex areas such as theoretical mathematics (Polu & Sutskever, 2020), undergraduate mathematics (Drori et al., 2022), and question answering with data retrieval (Sun et al., 2023b; Cheng et al., 2023).

PoT enhances performance due to the precision and verifiability inherent in code as a machine-executable language. Within task decomposition, it is not uncommon for the LLM to hallucinate incorrect subtasks and questions through CoT (Ji et al., 2023). PoT implementations from Chen et al. (2023b), Gao et al. (2023), and Ye et al. (2023b) show that by directly executing code and verifying outcomes post translation by LLMs, one can effectively mitigate the effects of incorrect reasoning in CoT. This is because the reasoning process must adhere to the logic and constraints explicitly specified by the program, thereby ensuring a more accurate and reliable outcome.

However, such improvements seen in the usage of code are not limited to purely executable coding languages such as Python or SQL, nor are they limited to tasks that are specifically rigid in structure such as mathematics (Drori et al., 2022) and data retrieval (Rajkumar et al., 2022). Enhancements also extend to the realm where even translating into pseudo-code to decompose a task can improve zero-shot performance (Lei & Deng, 2023) in word problems containing numbers, and general reasoning tasks such as StrategyQA (Geva et al., 2021).

### 3.3 Enable LLMs to Capture Structured Knowledge

**Code generation unveils superior structural commonsense reasoning.** Given that code possesses the graph structure of symbolic representations, translating textual graphs, tables, and images into code empowers a code-driven LLM to logically process such information according to code reasoning and generation principles.

Consequently, previous work (Madaan et al., 2022; Wang et al., 2023f) shows that LLMs undergoing extra pre-training on code may rival, or even exceed, their fine-tuned natural language counterparts in tasks involving structural commonsense reasoning, even with limited or no training data.

COCOGEN (Madaan et al., 2022) first reframed the commonsense reasoning graph completion task as a code generation task and demonstrated improved few-shot performance in reasoning graphs, table entity state tracking, and explanation graph generation.

Building on this perspective, CODE4STRUCT (Wang et al., 2023f) applied code-LLMs to semantic structures, focusing on the event argument extraction task. By leveraging code's features such as comments and type annotation, it achieved competitive performance with minimal training instances. Moreover, it surpassed baselines in zero-shot scenarios, benefiting from the inheritance feature and sibling event-type samples. ViStruct (Chen et al., 2023d) extended this approach further to multimodal tasks, leveraging programming language for representing visual structural information and curriculum learning for enhancing the model's understanding of visual structures.

**Markup code mastery evolves visually situated natural language understanding.** Another stream of research focuses on utilizing markup code such as HTML and CSS to delineate and derender structured graphical information in graphical user interfaces (GUIs) or visualizations such as plots and charts in documents. This markup code not only specifies content but also governs the layout of a web page, aiding large vision-language models (LVLMs) in capturing visually situated natural language (VSNL) information.

For LVLMs' markup code understanding, WebGUM (Furuta et al., 2023) exemplified autonomous web navigation. It employed a pre-training approach using webpage screenshots and the corresponding HTML as input, and navigation action as output. Outperforming SOTA, human, and other LLM-based agents, WebGUM showcased the effectiveness of pre-training model with markup code augmentation in webpage understanding.

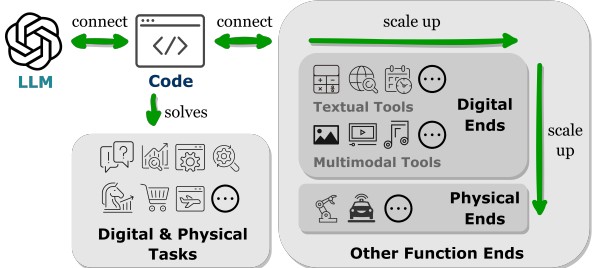

Figure 4: The code-centric tool-calling paradigm serves as a unified interface between LLMs and a large variety of functional ends, thus enabling many cross-modality and cross-domain tasks.

For markup code generation, pix2code (Beltramelli, 2017) and sketch2code (Robinson, 2019) pioneered machine learning methods to generate rendering code for GUIs or website mockups, in the pre-LLM era. Pix2Struct (Lee et al., 2023) achieved SOTA at that time in VSNL understanding by pre-training an image-to-text model on masked website screenshots, and further training with OCR, language modeling, and image captioning objectives. Building on this, MATCHA (Liu et al., 2023a) introduced additional pre-training objectives based on chart derendering, layout comprehension, and mathematical reasoning, surpassing SOTA on VSNL understanding tasks.

## 4 Code Connects LLMs to Other Function Ends

Table 1: Representative work connecting LLMs to different function ends for performing non-trivial tasks. Initial efforts embed tool calls rigidly within the LLMs' inference mechanism (indicated by "∗"), resulting in diminished flexibility and constrained tool accessibility. More recently, the *code-centric paradigm* establishes connections between LLMs and function ends through programming languages or pre-defined functions (indicated by "†"). This approach enhances the scalability of LLMs' function end invocation across diverse tools and execution modules.

| Major Type of Function Ends | Representative Work | Connecting Paradigm | Learning Method | Objectives or Problems to Solve |
|---|---|---|---|---|
| Single Tool | Retriever in REALM Guu et al. (2020) | Hardcoded in Inference Mechanism* | Example Fine-tuning | Augment LLMs with Tools |
| | Verifier in GSM8K Cobbe et al. (2021) | Hardcoded in Inference Mechanism* | Example Fine-tuning | |
| Limited Text-based Tools | Blenderbot3 Shuster et al. (2022) | Hardcoded in Inference Mechanism* | Example Fine-tuning | Open-domain Conversation |
| | LamDA Thoppilan et al. (2022) | Generate Pre-defined Functions† | Example Fine-tuning | |
| **Text-based Tools** | TALM Parisi et al. (2022b) | Generate Pre-defined Functions† | Iterative Self-play | Efficient and Generalizable Tool Using |
| | ToolFormer Schick et al. (2023) | Generate Pre-defined Functions† | Self-supervised Training | |
| **Multi-modal Modules** | MM-React Yang et al. (2023) | Generate Pre-defined Functions† | Zero-shot Prompting | Multi-modal Reasoning Tasks |
| | CodeVQA Subramanian et al. (2023) | Generate Python Functions† | Zero-shot & Few shot | |
| | VISPROG Gupta & Kembhavi (2023) | Generate Python Functions† | Zero-shot Prompting | |
| | ViperGPT Surís et al. (2023) | Generate Python Functions† | Zero-shot Prompting | |
| **Real-World APIs** | Code as Policies Liang et al. (2023a) | Generate Python Functions† | Few-shot Prompting | Better Robot Control |
| | Progprompt Singh et al. (2022) | Generate Python Functions† | Zero-shot Prompting | |
| | SayCan Ahn et al. (2022) | Generate Pre-defined Functions† | Zero-shot Prompting | |
| | RRR Cui et al. (2023a) | Generate Pre-defined Functions† | Zero-shot Prompting | Autonomous Driving Ecosystems |
| | Agent-Driver Mao et al. (2023) | Generate Pre-defined Functions† | Few-shot Prompting | |
| | LaMPilot Ma et al. (2023b) | Generate Python Functions† | Zero-shot & Few-shot | |

Recent studies show that connecting LLMs to other functional ends (i.e., augmenting LLMs with external tools and execution modules) helps LLMs to perform tasks more accurately and reliably (Mialon et al., 2023; Parisi et al., 2022a; Peng et al., 2023; Gou et al., 2023). These functional ends empower LLMs to access external knowledge, engage with diverse modalities, and interact effectively with various environments. As indicated in Table 1, we observe a prevalent trend where LLMs generate programming languages or utilize pre-defined functions to establish connections with other functional ends—a phenomenon we refer to as the *code-centric paradigm*.

In contrast to the rigid practice of strictly hardcoding tool calls within the LLMs' inference mechanism, the code-centric paradigm allows LLMs to dynamically generate tokens that invoke execution modules with adaptable parameters. This paradigm enables a simple and unambiguous way for LLMs to interact with other functional ends, enhancing the flexibility and scalability of their application. Importantly, as depicted in Figure 4, it allows LLMs to engage with numerous functional ends spanning diverse modalities and domains.

By expanding both the quantity and variety of functional ends accessible, LLMs can handle more complicated tasks.

In §4.1, we examine (digital) textual and multimodal tools connected to LLMs, while §4.2 focuses on physical-world functional ends, including robots and autonomous driving, showcasing the versatility of LLMs in tackling problems across various modalities and domains.

## 4.1 Relate LLMs to Digital Ends

**Text-Based Tools.** The code-centric framework has enhanced precision and clarity to LLMs' tool invocation, initially driving progress in text-based tools. Prior to the popularity of this code-centric paradigm, research on augmenting LMs with single tools like information retrivers (Guu et al., 2020; Lewis et al., 2020; Izacard et al., 2022; Borgeaud et al., 2022; Yasunaga et al., 2022) required a hardcoded-in-inference-mechanism (e.g. always calling a retriever before the generation starts), which was less flexible and harder to scale. TALM (Parisi et al., 2022b) first incorporates multiple text-based tools by invoking API calls with a pre-defined delimiter, enabling unambiguous calls to any text-based tools at any position of generation. Following their work, Toolformer (Schick et al., 2023) marks API calls with "<API> </API>" along with their enclosed contents. Later, diverse tool-learning approaches were introduced to facilitate the integration of numerous text-based tools across various foundational models (Song et al., 2023; Hao et al., 2023; Tang et al., 2023).

The code-centric framework facilitates the invocation of a diverse range of external text modules. These include calculators, calendars, machine translation systems, web navigation tools, as well as APIs from HuggingFace and TorchHub (Thoppilan et al., 2022; Yao et al., 2022c; Shuster et al., 2022; Jin et al., 2023; Yao et al., 2022a; Liu et al., 2023e; Jin et al., 2023; Patil et al., 2023).

**Multimodal Tools.** The high scalability of the code-centric LLM paradigm enables the extension of tool-learning to modalities other than text. Early work (Gupta & Kembhavi, 2023; Surís et al., 2023; Subramanian et al., 2023) use the code-centric paradigm to tackle the visual question answering task. For instance, VISPROG (Gupta & Kembhavi, 2023) curates various pretrained computer vision models and functions from existing image processing libraries (e.g. Pillow and OpenCV) as a set of APIs. The API calls can then be chained together as a program for question-targeted image understanding, where the program is generated via in-context learning with LLMs. Containing arithmetic in its API code language, the program is capable of performing simple arithmetic tasks, thus enabling VISPROG to answer concrete questions such as object counting. Similar work includes ViperGPT (Surís et al., 2023) and CodeVQA (Subramanian et al., 2023). Compared to VISPROG, they directly generate more flexible Python code using Codex. This enables them to potentially generate programs of more complex control flows using the pre-trained knowledge embedded in Codex. In addition to visual reasoning, code has also been used to connect LLMs with multi-modal generative tools in image generation tasks (Cho et al., 2023; Feng et al., 2023; Wu et al., 2023a), where code's unambiguous nature is leveraged in generating images that better match their text prompts.

Beyond the image-based tools, other modalities have been considered and used in a collaborative fashion by recent work (Shen et al., 2023; Yang et al., 2023; Liang et al., 2023d). For example, MM-REACT (Yang et al., 2023) considers video recognition models in their API, and Chameleon (Lu et al., 2023) includes tools such as visual text detector or web search. In HuggingGPT (Shen et al., 2023), the authors connect LLMs to various Hugging Face models and treat each model as an available API call. As a result, HuggingGPT is capable of performing an even wider range of tasks, such as audio-related tasks, that were previously unexplored. Pushing the API diversity further, TaskMatrix.AI (Liang et al., 2023d) uses a magnitude higher number of APIs, spanning from visual & figure APIs to music and game APIs. The flexibility of code facilitates LLMs to jointly use different multimodal tools. This makes LLMs more versatile and capable of acting as general-purpose multimodal problem solvers that can scale to many tasks.

## 4.2 Relate LLMs to Physical Ends

While the physical world offers a more immersive, contextually rich, and engaging interactive environment compared to the digital realm, the connection between LLMs and the physical world has been constrained until the advent of the code-centric paradigm. This paradigm allows for adaptable calls to tools and execution

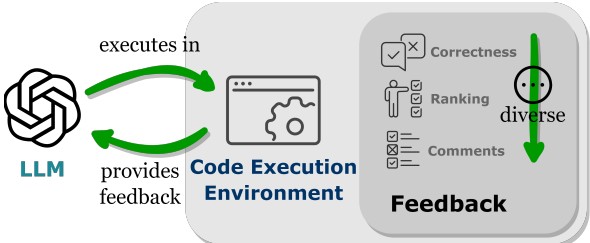

Figure 5: LLMs can be embedded into a code execution environment, where they collect faithful, automatic, and customizable feedback for self-improvement.

modules in the physical world, first sparking a wave of research exploring the integration of LLMs with robotics and autonomous driving.

One of the most successful approaches to employing LLMs to generate policy codes for real-world robotic tasks is PaLM-SayCan (Ahn et al., 2022), where LLMs comprehend high-level instructions and then call corresponding APIs for robotic control. Following SayCan, recent developments have shown that LLMs can serve as the brain for robotics planning and control through their powerful code generation capabilities. ProgPrompt (Singh et al., 2022), for instance, pioneered program-like specifications for robot task planning, while other researchers like Huang et al. (2023a), Liang et al. (2023a), and Vemprala et al. (2023) have extended this approach to a range of other tasks, including human-robot interaction and drone control.

Through code generation and tool learning, LLMs also show great potential in more complicated tasks such as human-vehicle interactions and autonomous driving (Cui et al., 2023b; Huang et al., 2023b; Li et al., 2023d). A prime tool learning example from the industry is Wayve's LINGO-1 (Wayve, 2023), which uses an open-loop vision, language, and action LLM to improve the explainability of driving models. Using instruction tuning, LLMs have advanced to the point where they can understand complex driving, navigation, and entertainment commands (Wang et al., 2023e), generate actionable codes (Ma et al., 2023b), and execute them by calling low-level vehicle planning and control APIs (Cui et al., 2023a; Sha et al., 2023; Mao et al., 2023).

Overall, despite challenges such as latency, accuracy issues, and the absence of adequate simulation environments, datasets, and benchmarks (Kannan et al., 2023; Chen & Huang, 2023; Cui et al., 2023b), LLMs show promise in understanding high-level instructions and executing code-related APIs in intricate domains like robotics and autonomous driving. Looking ahead, there's considerable potential for LLMs to bridge the gap between physical worlds and AI, influencing areas like transportation and smart manufacturing (Panchal & Wang, 2023; Zeng et al., 2023a).

## 5   Code Provides LLM with an Executable Environment for Automated Feedback

LLMs demonstrate performance beyond the parameters of their training, in part due to their ability to intake feedback, especially in real-world applications where environments are rarely static (Liu et al., 2023f; Wang et al., 2023d). However, feedback must be chosen carefully as noisy prompts can hinder LMs' performance on downstream tasks (Zheng & Saparov, 2023). Furthermore, as human effort is expensive, it is crucial that feedback can be automatically collected while staying faithful. Embedding LLMs into a code execution environment enables automated feedback that fulfills all of these criteria, as shown in Figure 5. As the code execution is largely deterministic, LLMs that intake feedback from the results of executed code remain faithful to the task at hand (Chen et al., 2023a; Fernandes et al., 2023; Scheurer et al., 2022). Furthermore, code interpreters provide an automatic pathway for LLMs to query internal feedback, eliminating the need for costly human annotations as seen when leveraging LLMs to debug or optimize faulty code (Chen et al., 2023a; Fernandes et al., 2023; Scheurer et al., 2022). In addition, code environments allow LLMs to incorporate diverse and comprehensive forms of external feedback, such as critic on binary correctness (Wang et al., 2023g), natural language explanations on results (Chen et al., 2023c), and ranking with reward values (Inala et al., 2022), enabling highly customizable methods to enhance performance.

We introduce the various types of feedback derived from code execution that can be jointly utilized to benefit LLMs in §5.1, and discuss common methods for utilizing this feedback to improve LLMs in §5.2.

## 5.1 Various Feedback from Code Execution

The code execution process enables assessing LLM-generated content with more comprehensive evaluation metrics derived from deterministic execution results, as opposed to relying solely on often ambiguous sequence-based metrics like BLEU (Papineni et al., 2002; Ren et al., 2020) and Rouge (Lin, 2004).

Straightforward methods for evaluating program execution outcomes and generating feedback include the creation of unit tests (Chen et al., 2021; Hendrycks et al., 2021; Austin et al., 2021; Li et al., 2022; Huang et al., 2022a; Lai et al., 2023) and the application of exact result matching techniques (Dong & Lapata, 2016; Zhong et al., 2017; Huang et al., 2022a). From these, feedback can be provided in two primary forms: simple correctness feedback and textual feedback. Simple feedback, indicating whether a program is correct or not, can be generated through critic models or rule-based methods (Wang et al., 2023g; Chen et al., 2023c).

For more detailed textual feedback, language models can be employed to produce explanations either about the program itself (Chen et al., 2023c), or to summarize comments on the program and its execution (Wang et al., 2023g; Chen et al., 2023c; Zhang et al., 2023a). Execution results can also be translated into reward functions using predefined rules. The rules map execution results into scalar values based on the severity of different error types, thus making the feedback format suitable for reinforcement learning approaches (Le et al., 2022). Moreover, additional feedback can be extracted by performing static analysis using software engineering tools. For instance, Wang et al. (2017) and Gupta et al. (2017) obtain extra information from the execution trace, including variable or state traces. Lai et al. (2023) demonstrates an effective way to extract extra feedback using surface-form constraints on function calls.

## 5.2 Methods for Enhancing LLM's Performance with Feedback

The feedback derived from code execution and external evaluation modules can enhance LLMs through three major approaches.

**Selection Based Method.** Selection-based methods, such as majority voting and re-ranking schemes, have proven effective in enhancing LLM performance in tasks such as code generation. These methods, originally developed for simple program synthesis, leverage code execution outcomes like the number of passed unit tests to choose the best-performing code snippet. Studies like Chen et al. (2018); Li et al. (2022) demonstrate the efficacy of majority voting, while Zhang et al. (2023b); Yin & Neubig (2019); Zeng et al. (2023b) showcase the advantages of re-ranking schemes. Building on this success, similar approaches have been adapted for more challenging tasks where code-LLMs are integrated in interactive environments, as shown in the work of Shi et al. (2022); Chen et al. (2022) for similar voting methods, and Ni et al. (2023); Inala et al. (2022); To et al. (2023) for re-ranking strategies. However, these approaches, while simple and effective, cause inefficiencies, as they necessitate multiple rounds of generation and the employment of additional re-ranking models to identify the optimal solution.

**Prompting Based Method.** Modern LLMs are equipped with the capability to reason in-context and directly integrate feedback from task descriptions into prompts, to certain extents. Improving LLM "self-debugging" with in-context learning prompts typically requires feedback presented as natural language explanations (Wang et al., 2023h; Chen et al., 2023c) or error messages derived from execution results, as these formats are more comprehensible for the LLM. This method is favored by most LLM-based agents (see §6) due to its automatic nature, computational efficiency, and lack of requirement for additional fine-tuning. However, the effectiveness of this approach heavily depends on the LLM's in-context learning capabilities.

**Finetuning Based Method.** In the aforementioned methods, neither the selection-based method nor the prompting-based method promises steady improvement over the task, as the LLMs' parameters remain unchanged. They require repeating the tuning process even when faced with similar problems. Finetuning approaches, on the other hand, fundamentally improve the LLMs by updating their parameterized knowledge.

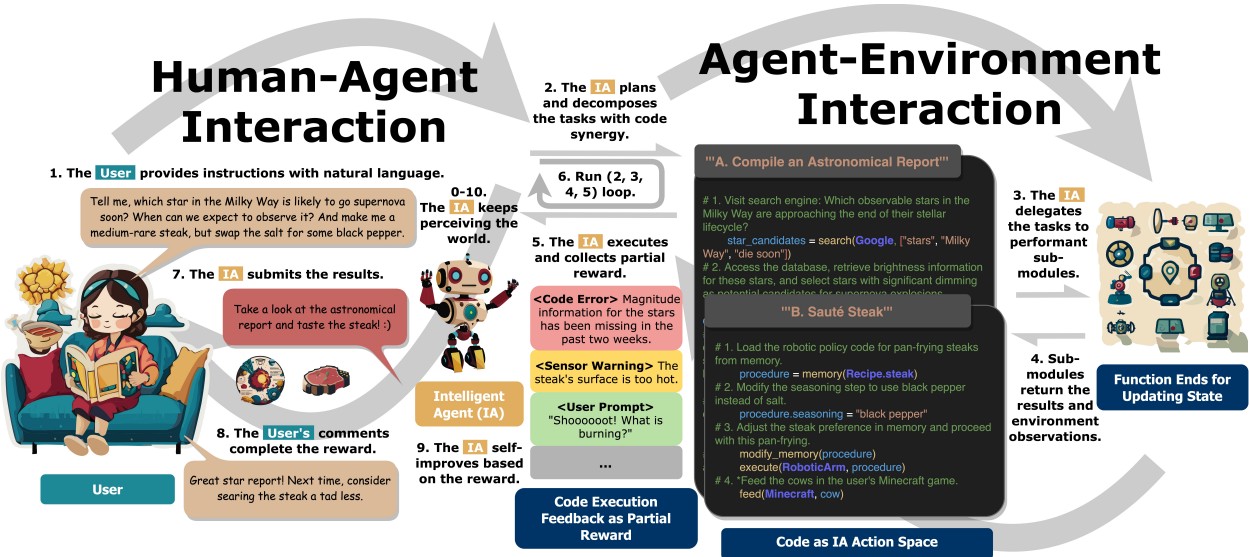

Figure 6: This figure illustrates the complete working pipeline of a LLM-based intelligent agent, mapping code-LLM abilities to specific phases: code-based planning in step (2), modular action parsing and tool creation in step (3), and automated feedback collection for enhanced agent self-improvement in step (5). Collectively, steps 0-10 in the entire loop benefit from code-LLMs' improved structured information understanding and perception.

Typical finetuning strategies include direct optimization, leveraging an external model for optimization, and training the model in a reinforcement learning paradigm. Wang et al. (2023g) exemplifies the direct optimization approach, where the original language model is fine-tuned with a feedback-conditioned objective. Haluptzok et al. (2022) presents a unique method where language models generate synthetic unit tests to identify and retain only correctly generated examples, which are then composed into correct question-answer pairs and employed to further fine-tune the models. CodeScore (Dong et al., 2023a) designs its loss function based on executability and the pass rate on the unit tests. For self-edit (Zhang et al., 2023a), it first wraps up execution results into textual comments, then trains an editor to further refine the program by accepting both the problematic program and the comments. Chen et al. (2023a) train a "refine model" which accepts the feedback and generated program first, then use the refined generated example to fine-tune the generation model, illustrating a layered approach to fine-tuning. CodeRL (Le et al., 2022) and Wang et al. (2022) apply reinforcement learning to improve the original language model. While Wang et al. (2022) aims at employing compiler feedback to obtain erroneous code, CodeRL (Le et al., 2022) empirically defines fixed reward values for different execution result types based on unit tests. Despite the discussed advantages, refining LLMs through finetuning typically involves a resource-intensive data collection process. Additionally, assessing predefined reward values, as exemplified in CodeRL (Le et al., 2022), poses certain challenges.

# 6 Application: Code-empowered LLMs Facilitate Intelligent Agents

In the preceding sections, our discussion highlighted the various ways in which code integration enhances LLMs. Going beyond, we discern that the benefits of code-empowered LLMs are especially pronounced in one key area: the development of IAs Xu et al. (2023). In this section, we underscore the unique capabilities bestowed upon these agents by code-empowered LLMs.

Figure 6 helps to illustrate a standard operational pipeline of an IA, specifically serving as an embodied general daily assistant. We observe that the improvements brought about by code training in LLMs are firmly rooted in their practical operational steps when serving as IAs. These steps include (i) enhancing the IA's decision-making in terms of environment perception and planning (§6.1), (ii) streamlining execution by grounding actions in modular and explicit action primitives and efficiently organizing memory (§6.2), and

(iii) optimizing performance through feedback automatically derived from the code execution environment (§6.3). Detailed explanations of each aspect will follow in the subsequent sections.

## 6.1 Decision Making

**Environment Perception**   As depicted in Figure 6 at step (0-10), the IA continuously perceives the world, engaging in interactions with humans and the environment, responding to relevant stimuli (e.g., human instructions for meal preparation), and planning and executing actions based on the observed environmental conditions (e.g., the kitchen layout). Utilizing LLMs as decision centers for IAs requires translating environmental observations into text, such as tasks based in the virtual household or Minecraft (Shridhar et al., 2020; Côté et al., 2018; Wang et al., 2023b; Zhu et al., 2023). The perceived information needs to be organized in a highly structured format, ensuring that stimuli occurring at the same moment (e.g., coexisting multimodality stimuli) influence the IA's perception and decision simultaneously without temporal differences, a requirement that contrasts with the sequential nature of free-form text. Through pre-training on code, LLMs acquire the ability to better comprehend and generate structured representations resembling class definitions in code. This structured format, where class attributes and functions are permutation invariant, facilitates agents in perceiving structured environmental observations during task execution.

One such intuitive example is web-page-based environments which are highly structured around HTML code. In agent tasks like web shopping (Yao et al., 2022b), web browsing (Deng et al., 2023b), and web-based QA (Nakano et al., 2021; Liu et al., 2023d), it is preferred to translate the web-based environment into HTML code rather than natural language, directly encompassing its structural information and thereby improving the LLM agent's overall perception (Yang et al., 2024). Moreover, in robotics research by Singh et al. (2022) and Liang et al. (2023a), the IAs are prompted with program-like specifications for objects in the environment, enabling the LLM to generate situated task plans based on the virtual objects they perceived.

**Planning**   As illustrated in Figure 6 at step (2), IAs must break down intricate tasks into finer, manageable steps. Leveraging the synergized planning abilities of code-LLMs, IAs can generate organized reasoning steps using modular and unambiguous code alongside expressive natural language. As discussed in §2.2, when code-LLMs are employed for planning agent tasks, they exhibit enhanced reasoning capabilities. In addition, they generate the sub-tasks as executable programs when necessary, yielding more robust intermediate results, which the IA conditions on and refines its planning with greater precision. Furthermore, the IA seamlessly integrates performant tool APIs into planning, addressing the limitations such as poor mathematical reasoning and outdated information updates faced by vanilla LLMs during planning.

Typical examples that utilize code for planning are in two main categories. Progprompt (Singh et al., 2022) and Code as Policies (Liang et al., 2023a) represent the work utilizing code for better robot control. Both work highlight the benefits brought by code-based planning as they not only enable direct expressions of feedback loops, functions, and primitive APIs, but also facilitate direct access to third-party libraries. Another stream of work is concerned with the scenario when the agents' programming and mathematical reasoning abilities are crucial, like solving maths-related problems (Gao et al., 2023; Wang et al., 2023h) or doing experiments in the scientific domain (Boiko et al., 2023; Liffiton et al., 2023).

## 6.2 Execution

**Action Grounding**   As depicted in Figure 6 at step (3), when the IA interfaces with external function ends according to the planning, it must invoke action primitives from a pre-defined set of actions (i.e., functions). Given that modern LLMs are trained in formal language and can generate highly formalized primitives, the IA's generation can be directly parsed and executed, eliminating the necessity for additional action primitive grounding modules.

Connecting the IA with other function ends requires grounding actions into formalized function-like primitives. For instance, in a benchmark evaluating LLMs as agents in real-world scenarios (Liu et al., 2023f), seven out of eight scenarios involve code as the action space.

Previous work generating agent plans with pure natural language necessitate an additional step-to-primitive module to ground those planning steps into code (Wang et al., 2023c; Yin et al., 2023). In contrast, IAs that plan with code-LLMs generate atomic action programs (Yao et al., 2022d; Wang et al., 2023h; Liang et al., 2023a; Singh et al., 2022), and can have their generation quickly parsed for execution.

**Memory Organization**    As depicted in Figure 6 at step (3) and the component labeled "Function Ends for Updating State," the IA typically necessitates an external memory organization module to manage exposed information (Wang et al., 2023d), including original planning, task progress, execution history, available tool set, acquired skills, augmented knowledge, and users' early feedback. In this context, Code-LLM aids the IA's memory organization by employing highly abstract and modular code to record, organize, and access memory, especially for expanding the available tool set and manage acquired skills.

Typically, agent-written code snippets can serve as parts of the toolset, integrated into the memory organization of agents. This stream of research is known as tool creation approaches. TALM (Cai et al., 2023) proposes to use stronger agents (e.g. GPT-4 based agents) to write code as part of memory for weaker agents (e.g. GPT-3.5 based agents). In Creator (Qian et al., 2023b), agents themselves are highlighted as not only users of the tools but also their creators. They proposed a four-stage tool-creation framework that enables agents to write code as part of their executable tool set. Going further, Craft (Yuan et al., 2023) focuses on ensuring the created tools are indeed executable, making the framework more robust. Another work sharing this idea is Voyager (Wang et al., 2023b), in which the agent store learned skills in code format and execute them in the future when faced with similar tasks.

## 6.3   Self-improvement

As illustrated in Figure 6 at step (5), when the IA's decision center, i.e., the LLM, operates within a code execution environment, the environment can integrate various evaluation modules to offer automated feedback (e.g., correctness, ranking, detailed comments). This significantly enhances the IA's early error correction and facilitates self-improvement.

Voyager (Wang et al., 2023b) is a good example for agents that use feedback from the simulated environment. The agent learns from failure task cases and further horn its skills in Minecraft. Chameleon (Lu et al., 2023) receives feedback from a program verifier to decide whether it should regenerate an appropriate program. Mint (Wang et al., 2023h) can receive feedback from proxies, and the agent can thus self-improve in a multi-turn interactive setting. Importantly, this ability to self-improve from execution feedback is fundamental for agents' success at solving scientific problems (Bran et al., 2023; Swan et al., 2023; Wu et al., 2023c).

# 7   Challenges

We identify several intriguing and promising avenues for future research.

## 7.1   The Causality between Code Pre-training and LLMs' Reasoning Enhancement

Although we have categorized the most pertinent work in §3.2, a noticeable gap persists in providing explicit experimental evidence that directly indicates the enhancement of LLMs' reasoning abilities through the acquisition of specific code properties. While we intuitively acknowledge that certain code properties likely contribute to LLMs' reasoning capabilities, the precise extent of their influence on enhancing reasoning skills remains ambiguous. In the future research endeavors, it is important to investigate whether reinforcing these code properties within training data could indeed augment the reasoning capabilities of trained LLMs. If it is indeed the case, that pre-training on specific properties of code directly improves LLMs' reasoning abilities, understanding this phenomenon will be key to further improving the complex reasoning capabilities of current models.

## 7.2 Acquisition of Reasoning Beyond Code

Despite the enhancement in reasoning achieved by pre-training on code, foundational models still lack the human-like reasoning abilities expected from a truly generalized artificial intelligence. Importantly, beyond code, a wealth of other textual data sources holds the potential to bolster LLM reasoning abilities, where the intrinsic characteristics of code, such as its lack of ambiguity, executability, and logical sequential structure, offer guiding principles for the collection or creation of these datasets. However, if we stick to the paradigm of training language models on large corpora with the language modeling objective, it's hard to envision a sequentially readable language that is more abstract, highly structured, and closely aligned with symbolic language than formal languages, exemplified by code, which are prevalent in a substantial digital context. We envision that exploring alternative data modalities, diverse training objectives, and novel architectures would present additional opportunities to further enhance the reasoning capabilities of these models.

## 7.3 Challenges of Applying Code-centric Paradigm

The primary challenge in LLMs using code to connect to different function ends is learning the correct invocation of numerous functions, including selecting the right function end and passing the correct parameters at an appropriate time. Even for simple tasks like simplified web navigation with a limited set of action primitives like mouse movements, clicks, and page scrolls, few shot examples together with a strong underlying LLM are often required for the LLM to precisely grasp the usage of these primitives Sridhar et al. (2023). For more complex tasks in data-intensive fields like chemistry Bran et al. (2023), biology, and astronomy, which involve domain-specific Python libraries with diverse functions and intricate calls, enhancing LLMs' capability of learning the correct invocation of these functions is a prospective direction, empowering LLMs to act as IAs performing expert-level tasks in fine-grained domains.

## 7.4 Learning from multi-turn interactions and feedback

LLMs often require multiple interactions with the user and the environment, continuously correcting themselves to improve intricate task completion Li et al. (2023c). While code execution offers reliable and customizable feedback, a perfect method to fully leverage this feedback has yet to be established. As discussed in § 5.2, we observed that selection-based methods, though useful, do not guarantee improved performance and can be inefficient. Prompting-based methods heavily depend on the in-context learning abilities of the LLM, which might limit their applicability. Fine-tuning methods show consistent improvement, but data collection and fine-tuning are resource-intensive and thus prohibitive. We hypothesize that reinforcement learning could be a more effective approach for utilizing feedback and improving LLMs. This method can potentially address the limitations of current techniques by providing a dynamic way to adapt to feedback through well-designed reward functions. However, significant research is still needed to understand how reward functions should be designed and how reinforcement learning can be optimally integrated with LLMs for complex task completion.

# 8 Conclusion

In this survey, we compile literature that elucidates how code empowers LLMs, as well as where code assists LLMs to serve as IAs. To begin with, code possesses natural language's sequential readability while also embodying the abstraction and graph structure of symbolic representations, rendering it a conduit for knowledge perception and reasoning as an integral part of the LLMs' training corpus based on the mere language modeling objective. Through a comprehensive literature review, we observe that after code training, LLMs *i)* improve their programming skills and reasoning, *ii)* could generate highly formalized functions, enabling flexible connections to diverse functional ends across modalities and domains, and *iii)* engage in interaction with evaluation modules integrated in the code execution environment for automated self-improvement. Moreover, we find that the LLMs' capability enhancement brought by code training benefits their downstream application as IAs, manifesting in the specific operational steps of the IAs' workflow regarding decision-making, execution, and self-improvement. Beyond reviewing prior research, we put forth several challenges in this field to serve as guiding factors for potential future directions.

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

## A Discussions

### A.1 Intrinsic Qualities of Code that Contribute to LLM Empowerment

Reflecting on our definition of code in the introduction section (§1) as formal languages that are both human-interpretable and machine-executable, we highlight that while some features are shared by all code, programming language, as the most well-known and most established type of code, enjoy some unique advantages. In Figure 7, we provide a case study comparing code and natural language.

First, we talk about the core feature shared by all code within the range of our definition. The inherent nature of code is that they are explicit and have clear definitions for every single line, while natural language is generally in free form and can be very ambiguous. Consequently, code is significantly better at expressing detailed commands, signifying a specific step, and transmitting control signals. This generally led to the improvement in §4, the improvement for more controlled planning (cf. planning part in §6.1), and also helped with action execution (§6.2).

Programming languages, a critical component of the code family, are specifically designed for machine communication. Their advantages extend beyond mere explicitness and clarity. One overwhelming feature of programming languages (though some formal languages also define logical commands and loops) is that they contain structural definitions. Some well-known features are logical operands (If & Else), loops (For & While), nesting (within Functions), and even class definition and class inheritance (Object Oriented Programming). This feature makes them super suitable for expressing nesting and complicated structures (cf. §3.3 and the perception part in §6.1). Another feature is that programming languages are often paired with a very powerful execution environment. This executable feature benefits much as it naturally delegates some harder tasks to lower level, like arithmetic computing or interacting with a simulated environment when connecting to a Database, Minecraft, and so on, also facilitating reasoning discussed in §3.2. What's more, the execution often includes feedback mechanisms, which can be valuable for further refining the generator (§5 and §6.3).

### A.2 Breadth by Code Delegation or Depth by Multimodality Joint Learning

LLMs can swiftly and cost-effectively address tasks involving more data modalities by utilizing code to invoke tools. Simultaneously, joint fine-tuning on multimodal data enhances the model's precision and robustness in perceiving each modality, resulting in superior task performance. For instance, on the VQA dataset GQA (Hudson & Manning, 2019), ViperGPT (Surís et al., 2023), a typical code-centric paradigm, marginally surpasses the multimodal model BLIP-2 (Li et al., 2023b) in the zero-shot scenario after learning visual model API usages. However, its accuracy remains significantly lower than other supervised multimodal models. It is also still uncertain whether this approach will surpass the state-of-the-art models on multi-modal procedural

```
class IntelligentAgent
    """
    An intelligent agent utilizing a decision center (default: LLM) and a toolbox of available tools.

    Parameters:
    - decision_center: The decision-making center for the agent, defaults to an LLM.
    - toolbox: A list of available tools for the agent, default includes GOOGLE, Minecraft, and RoboticArm.

    Methods:
    - cs_rookie_ritual(): Executes a rookie ritual for a computer science rookie, obtaining plans from the decision
center and performing actions with the specified tools.
    """
    def __init__(self, decision_centor=LLM, toolbox=[GOOGLE, Minecraft, RoboticArm]):
        """
        Initialize an IntelligentAgent instance.

        Args:
        - decision_center: The decision-making center for the agent, defaults to an LLM.
        - toolbox: A list of available tools for the agent, default includes GOOGLE, Minecraft, and RoboticArm.
        """
        self.decision_centor = decision_centor          1. Object-Oriented Programming
        self.toolbox = toobox                              Adv: Structured

    def cs_rookie_ritual(self):                 2. Functional Programming
        """                                        Adv: Modular & Explicit
        Execute a rookie ritual for a computer science rookie using the decision center and specified tools.
        """
        plans = self.decision_centor("Hello, World!", toolbox=self.toolbox)
        for tool, action in plans:                  3. Procedural Programming
            action(tool)                               Adv: Step-by-Step
```

Figure 7: We generate pseudo-code for the "IntelligentAgent" class and employ ChatGPT to compile its docstring. By contrasting the self-explanatory code with its natural language docstring, we observe that code exhibits greater structure, expressiveness, and logical coherence, underscoring certain advantages of code over natural language.

planning (Liu et al., 2023b). One reason is that the code-centric paradigm's effectiveness hinges on the central decision model and individual task execution components. This makes code-delegation approaches susceptible to error accumulation across steps and highly influenced by the worst-performing sub-modules or tools. Nevertheless, code delegation remains essential, as certain tools' advantages, such as the precision of calculators and the flexibility of search engines, cannot be learned by training multimodality models alone. The high extensibility of the code-centric paradigm to various tools and modalities also makes it a perfect fit for domains where training data is hard to collect at scale. We anticipate that the central decision model, utilizing code to invoke tools, will evolve from text-only LLMs to multimodality models capable of comprehensively understanding and processing multimodal data.

## A.3 The Potential of Using Code-centric Framework for Intelligent Agent Construction

We observed a rising trend in leveraging code in the construction of LLM-based intelligent agents. As shown in §6, we showed three major scenarios where agents can effectively benefit from code usage. We also identified that this trend mainly originated from the increasing need to evaluate agents in a real-world scenario, where executive environments and interactions are everywhere. A natural question arises: Does code have the potential to substitute natural language and become the dominant media in the construction of agents?

A lot of work has begun to adopt this approach, like Voyager (Wang et al., 2023b) in a simulated Minecraft environment. They used code for high-level planning, low-level control sequence, and execution to interact

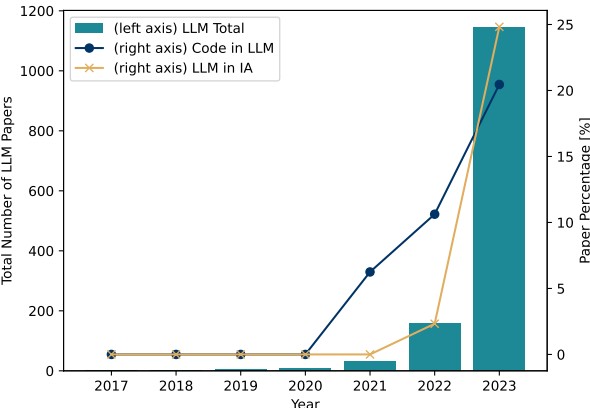

Figure 8: Paper statistics from Arxiv. We identified a significant and growing trend in recent research focused on code-based large language models (LLMs) and LLM-based intelligent agents (IAs). Code usage contributes much to the success of these cutting-edge models and systems.

with the environment. Acquired skills are also organized in the format of code snippets. With the code-centric paradigm, the framework is highly automatic and efficient. However, it's also true that many framework today are still using natural language for planning, probably because they provide more human-interpretable reasoning steps. Human feedback in natural language is also widely used to harvest strong reward models that reflect real human preferences. We hypothesize that the integration of code will continue gaining popularity on our path to AGI, especially for facilitating interactions between agents and the real world. Nevertheless, natural language could hardly be replaced regarding the interaction between agents and humans.(Drori et al., 2022; Chen et al., 2023b; Lei & Deng, 2023). Leveraging this understanding, we aim to explore novel research avenues in LLM reasoning inspired by the utilization of "code".

## B    Paper Statistics from Arxiv

We write a Python script that serves as a web scraper to extract paper details from the ArXiv preprint server, specifically focusing on the field of computer science. The web scraper gathers information about papers related to specific topics, including code, LLM, and IA. The script navigates through the ArXiv website, fetching essential details such as paper title, abstract, authors, and subject categories. We analyze and visualize data related to these papers in Figure 8, intending to provide insights into the trends and relationships between LLMs, code-related topics, and IAs in the past few years.

## C    Benchmarks for Evaluating Complex Reasoning with Code:

While there exist many benchmarks used to evaluate the abilities of LLMs (Liang et al., 2023c) across many disciplines, the benchmarks that most directly evaluate LLMs pre-trained on code in complex reasoning tasks are programming benchmarks such as CodeBLEU (Ren et al., 2020), where metrics that better match a human's evaluation of what is good logical, interpretable, and syntactically concise code was created, and CodeXGLUE (Lu et al., 2021) where multiple programming tasks such as code repair and code defect detection were accumulated into one dataset. Other suitable benchmarks include math datasets such as many of MIT's undergraduate math courses such as calculus and linear algebra, (Drori et al., 2022), LILA, a compilation of 23 tasks that test for mathematical abilities, language format, language diversity, and external knowledge abilities of LLMs (Mishra et al., 2023), and theorem proving from the metamath theorem code language (Polu & Sutskever, 2020). Others include question-answering tasks that require complex abilities to perform data retrieval in SQL databases (Ye et al., 2023b), such as those seen by the Spider dataset (Yu et al., 2019; Rajkumar et al., 2022).

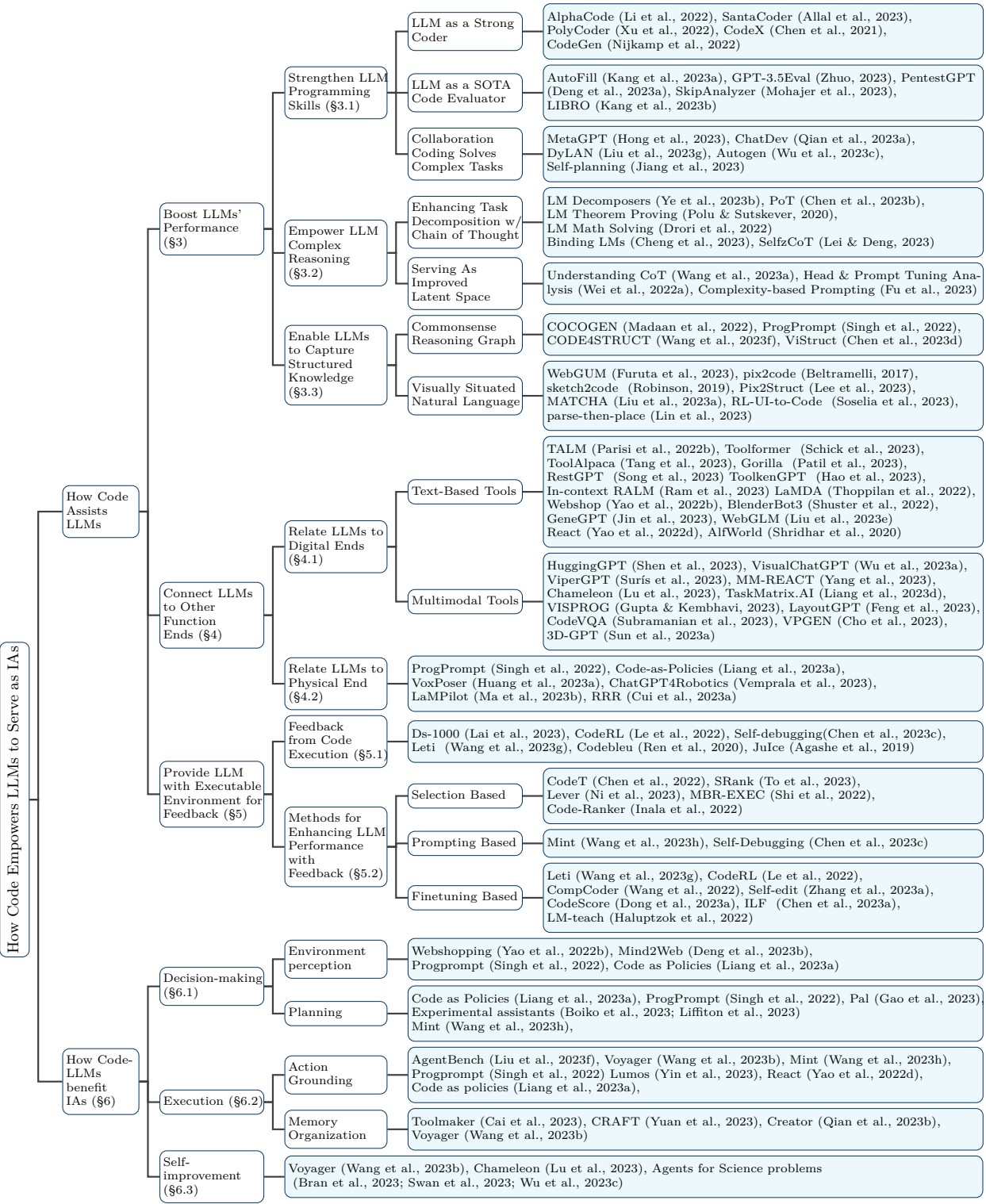

Figure 9: We hereby provide a complete list of the papers included in our survey.

| Major Functionalities Facilitated | Key Features of Code |
|---|---|
| **Strengthen LLMs' Programming Skills** Correspond to §3.1 and Figure 3 (a) | **Machine Executable:** Pretraining with Code, the LLM is able to write code, evaluate code, and utilize collaborative coding to solve complex tasks. |
| **Empower LLMs' Complex Reasoning** Correspond to §3.2 and Figure 3 (b) | **Structured and Expressive:** The step-by-step nature of code benefits CoT. **Machine Executable:** LLMs can utilize code to help with certain capabilities like mathematical reasoning. |
| **Enhance LLMs' Structured Knowledge** Correspond to §3.3 and Figure 3 (c) | **Structured and Expressive:** Code can be used to express complex structures, some programming language features like logical expressions and class inheritance will be especially useful. |
| **Connect LLMs to Other Functional Ends** Correspond to §4 and Figure 4 | **Explicit and Unambiguous**: Code is more explicit and clear than natural language, thus can better express clear instructions of connecting to any other functional ends. |
| **Provide LLMs w/ Environmental Feedback** Correspond to §5 and Figure 5 | **Machine Executable**: Code execution result can be treated as feedback to finetune the LLMs further and make their performance more desirable |
| **Help with IAs' Decision-Making** Correspond to §6.1 and Figure 6 | **Structured and Expressive**: Code pretraining Enhances agents' ability to precept structural knowledge, step by step feature improves CoT planning, logical expressions and nesting help with better control flow. **Machine Executable**: Help with solving mathematical tasks. |
| **Help with IAs' Action Execution** Correspond to §6.2 and Figure 6 | **Explicit and Unambiguous:** Unambiguous function calling help with action grounding. **Machine Executable:** Agents can write reusable code snippets as its memory |
| **Help with IAs' Self-improving** Correspond to §6.3 and Figure 6 | **Machine Executable:** Agent code execution results reflect potential environment change and can be utilized for self-improving |

Table 2: We conclude the three major key features of code and correspond them to the major functionalities of LLMs and IAs they facilitated. The three key features are namely **Structured and Expressive**, **Machine Executable** and **Explicit and Unambiguous**. More details of how these features assist LLMs and IAs can be found in the preamble of each section.

## D   The Comprehensive Paper List

To complement the core paper list presented in Figure 2, we have included a comprehensive list of papers in Figure 9. It is important to note that this list excludes papers used for performance comparisons between code and natural language. Instead, it focuses on papers that have utilized code to augment the capabilities of Large Language Models and intelligent agents.

## E   Mappings of Sections to Core Code features

In each section, we identify key code features that contribute to the success of enhancing Large Language Models and Intelligent Agents. The correlation between each section and its core features is detailed in Table 2. We have classified code features into three main categories: Machine Executable, Structured and Expressive, and Explicit and Unambiguous. Various aspects of these core features play a pivotal role in the effective use of code. Detailed explanations of these aspects are provided in the right column of the table. Additionally, further information can be found in the preamble of each respective section.

