# OpenReview forum: "If LLM Is the Wizard, Then Code Is the Wand: A Survey on How Code Empowers Large Language Models to Serve as Intelligent Agents"
_TMLR — Withdrawn by Authors_

### Review · Reviewer_1zXc · 2025-04-20

**Summary Of Contributions:**

The paper provides the first comprehensive survey of how integrating code into large language model (LLM) pre‑training and fine‑tuning not only enhances code‑generation capabilities but also unlocks complex reasoning, enables tool invocation, provides automated feedback loops, and maps to intelligent‑agent workflows. The survey also traces how these code‑empowered capabilities concretely benefit IAs in perception, planning, action grounding, memory organization, and continual self‑improvement. It finally identifies four open challenges—establishing causality between code and reasoning, extending reasoning beyond code, scaling function‑invocation learning, and leveraging multi‑turn feedback—that chart promising research directions.

**Audience:**

Yes

**Broader Impact Concerns:**

- Bias and fairness. Pre‑training on public code repositories may propagate licensing conflicts, insecure coding patterns, and biases in downstream IA decisions.

- Privacy and security. Automated code generation risks leaking sensitive information (e.g. API keys) and introducing vulnerabilities.

**Claims And Evidence:**

Yes

**Requested Changes:**

- Add a “Contributions” bullet list. Explicitly enumerate the contributions in the Introduction.

- Include quantitative meta‑analysis. Summarize key metrics in a table or chart.

- Add a Broader Impact section. Discuss potential societal, security, and privacy risks of autonomous code‑enabled agents and propose mitigations.

**Strengths And Weaknesses:**

Strengths

- Comprehensive taxonomy.
Organizes ∼150 works into a clear structure spanning programming skills, reasoning, tool‑centred execution, and agent workflows.

- Depth across modalities.
Covers digital APIs, multimodal tools and physical‐world control, illustrating broad applicability .

- Executable feedback focus.
Highlights how code execution affords faithful, automated signals for selection‑, prompting‑, and fine‑tuning‑based improvements.

- Concrete agent perspective.
Connects the survey’s insights directly to IA operational steps, offering practitioners actionable guidance.

Weaknesses
- Limited empirical comparison. Lacks quantitative meta‑analysis (e.g. performance gains across models/tasks) that could strengthen claims.

- Broader impact omission. No dedicated section on ethical implications or risk mitigation for code‑empowered LLM agents

---

### Review · Reviewer_HMEo · 2025-05-13

**Summary Of Contributions:**

The authors survey several works that look at how code and LLMs interact. The majority of focus is on how code itself enables better LLM capabilities, at various stages of the LLM usage pipeline (pre-training, fine tuning, inference, etc). A final portion is dedicated to how this improves things in an Intelligent Agent setting.

**Audience:**

Yes

**Broader Impact Concerns:**

I don’t think there is any broader impact implications that arise from this work.

**Claims And Evidence:**

Yes

**Requested Changes:**

Aside from the core criticisms raised in the “Weaknesses” response, I would recommend making the following changes:

- List the procedure you used to find+include papers for the survey. How did you initially gather them all? What filters did you use for inclusion? Etc.

- The “magician” rhetoric is a bit over the top, and not used so extensively. I’d recommend removing it from the title and caption of Figure 1. It adds little to the paper.

- At the start of 2.2, you include a formula with 0 context or explanation. I think I have an understanding of what the symbols mean, but you should detail this in the text.

- On page 13, you say “further horn its skills”. I think you mean “hone”

- Appendix B seems mostly an aside, and should probably be removed.

**Strengths And Weaknesses:**

The paper is well written, and the ontology breakdown is extremely compelling. Figure 2 highlights the scope of the work, and the level of detail is fairly decent throughout. Another strength of the work is the set of key challenges that remain. While they could be more concrete in what’s to be done, they serve as an excellent starting point for anyone looking to pursue further work in one of the several areas identified.

One weakness is the lack of capability understanding. A breadth of areas is covered, but it’s unclear just how capable each of the state-of-the-art methods are for that area. The descriptions are fairly surface-level, which is a bit of a weakness, but given the breadth of work covered, I’m not sure I would recommend diving deeper into the landmark papers. There is merit in the breadth presented.

Another weakness is the lack of connection to “human interpretability”. You make the (important) point that “code” is defined to be the languages that are machine *and* human readable. But what about the situation where you have code produced by LLMs that human’s interface with? Modifications, verifications, etc. There is an entire element of mixed initiative opportunity between LLMs and humans, on the common shared understanding of code, that seems to be lacking from the work.

Finally, one noticeable gap is in the description of planning approaches. For challenging multi-step planning problems, it has been well-established that LLMs (even the “reasoning” versions like R1 or o1) are poor at producing plans in novel settings. Conversely, using LLMs to generate code that planners can use (I.e., PDDL) is showing signs of promise. You can look up the work of Karthik Valmeekam (or more broadly, that of Subbarao Kambhampati) to see several works along this line. Also, [this survey](https://arxiv.org/abs/2503.18971) provides a reasonable summary of the area. Additionally, [this](https://arxiv.org/abs/2503.18809) is one recent work that is *very* relevant to a lot of the claims being made in this survey.

---

### Review · Reviewer_Awta · 2025-06-02

**Summary Of Contributions:**

The survey paper explains the widespread adoption of code-specific training in the general LLM training paradigm and how code enhances LLMs to act as Intelligent Agents. The survey paper claims that training with code (a) Strengthens LLMs’ programming and
code evaluation skills, (b) Empower LLMs’ complex reasoning skills, and (c) Enables LLM to more accurately model structured knowledge.

**Audience:**

Yes

**Broader Impact Concerns:**

No broader impact concern.

**Claims And Evidence:**

Yes

**Requested Changes:**

It would be good to incorporate quantitative data to support the main assertions in the paper (e.g., improved reasoning from code data).

The causal relationship behind some of the observations such as code training improving reasoning is not obvious and it would be good to include synthesis of existing work on this along with some counterexample/critique of evidences that do not support the central narrative (if any). This will present a balanced yet informative view expected of a survey.

**Strengths And Weaknesses:**

The paper identifies several advantages of training with code: enhances reasoning and programming ability, enables structured, executable output and integration with external tools and facilitates feedback loops through executable environments with improved reasoning.

The paper defines the "code-centric tool-calling paradigm" which allows LLMs to dynamically interface with APIs, digital tools, robotics, and multimodal systems using generated code as a bridge. By embedding LLMs into code execution environments, they can receive automated, structured feedback (e.g., unit test results, program correctness, static analysis), which helps improve model outputs over time. The paper identifies three feedback utilization strategies: selection-based, prompting-based, and fine-tuning-based methods.

Several applications of Code-LLMs as Intelligent Agents are identified. The include perception and structured reasoning (e.g., web agents reading HTML), planning and execution using modular, formalized primitives, tool creation and memory organization (e.g., agents writing and storing tools in code), and autonomous self-improvement by interpreting execution results and refining their actions.

Weaknesses:

- Even though the paper is a survey paper, the strong claim of code being a wand merits some empirical evaluation to backup the claims in the paper quantitatively. Including experimental results from prior papers could also suffice.

- The paper reads more as a compendium of observations from different papers and lacks significant synthesis.

---

### Note · Authors · 2025-06-14

I have read and agree with the venue's withdrawal policy on behalf of myself and my co-authors.